# Sensory Integration for Postural Control in Rheumatoid Arthritis Revealed by Computerized Dynamic Posturography

**DOI:** 10.3390/ijerph20064702

**Published:** 2023-03-07

**Authors:** Marta Cristina Rodrigues da Silva, Deyse Borges Machado, Luis Mochizuki, Melissa Andrea Jeannet Michaelsen Cardoso, Juliane de Oliveira, Monique da Silva Gevaerd, Ulysses Fernandes Ervilha, Monique Oliveira Baptista Cajueiro, Susana Cristina Domenech

**Affiliations:** 1College of Health and Sport Sciences, State University of Santa Catarina, Florianópolis 88080-350, Brazil; 2School of Arts, Sciences and Humanities, University of São Paulo, São Paulo 03828-000, Brazil; 3College of Distance Learning, State University of Santa Catarina, Florianópolis 88035-001, Brazil

**Keywords:** rheumatoid arthritis, postural control, balance control

## Abstract

Rheumatoid arthritis (RA) is a systemic autoimmune disease that impairs mobility. How does sensory information influence postural responses in people with RA? The aim of this study was to evaluate the postural control of people with RA during a sensory organization test, comparing how sensory information influences postural responses in people with rheumatoid arthritis compared with healthy people. Participants were 28 women with rheumatoid arthritis (RA group) and 16 women without any rheumatoid disease (Control group CG). The Sensory Organization Test (SOT) was performed on a Smart Balance Master^®^ (NeuroCom International, Inc., Clackamas, OR, USA) and center of pressure (COP) was measured. SOT conditions: SOT1 (eyes open, fixed support surface and surround; SOT2) eyes closed, fixed support surface and surround; and SOT5) eyes closed, sway-referenced support surface, and fixed surround. To compare the demographic and clinical aspects between groups, independent *t*-test or Mann–Whitney’s U-test were used. Differences were found between groups. Between SOT conditions, for CG and RA, COP was faster for SOT-5 than SOT-1, while SOT-1 and SOT-2 presented similar COP velocity. For SOT-2 and SOT-5, COP was larger for the RA group. For both groups, SOT-1 presented the smallest COP, and SOT-5 showed the largest COP.

## 1. Introduction

Rheumatoid arthritis (RA) affects about 1% of the world’s population. This systemic autoimmune disease affects the joint synovial membrane causing inflammation and pain, weakness, less joint mobility, and decreased proprioception [1]. These issues modify the functional capacity [2,3] and impair how postural control will act [4] during activities of daily living (ADL), increasing the risk of fall [5,6]. Postural responses in RA are poorly studied, while the sagittal balance is impaired by such systemic disease [7].

Changes in postural dynamic responses can be associated with impairments in motor control. Sensory Organization Test (SOT) quantifies what happens with the postural response during quiet standing [4,5,6,7,8], measuring the postural sway when one sensory source of information is missing or conflicting [9]. Thus, the compensatory capacity to adjust the importance of a sensory information source can be evaluated [10]. People with RA have a worse static postural balance performance with visual restriction or during unipodal support compared to healthy peers [5,11,12,13,14]. These findings raised the following question: How do people with RA control balance with restricted sensory information? Therefore, this study aimed to evaluate the postural control of people with RA during a Sensory Organization Test, comparing how sensory information influences postural responses in people with RA and matched healthy peers. The hypothesis of the study is that the postural sway of people with RA will be larger than for those who do not have this disease.

## 2. Materials and Methods

This is a cross-sectional study. Procedures were approved by the Local Ethical Committee (protocol 70650417.0.0000.0118), which is in accordance with the Declaration of Helsinki. All participants gave written informed consent to participate in the study.

### 2.1. Participant Recruitment and Allocation

Twenty-eight women diagnosed with rheumatoid arthritis (RA group) according to the 1987 American College of Rheumatology criteria [15], recruited from public and private health clinics in Florianópolis, Brazil; and 16 non-RA women (Control group, CG), matched by age (±8 years) were included. Inclusion criteria were the absence of unstable heart condition, chronic obstructive pulmonary disease, neurological diseases, or cancer, and the absence of any orthopedic condition that could impair unassisted walking or to execute functional tests. People with any lower limb joint health condition (or using lower limb prosthetics), or those with altered cognitive state were excluded.

### 2.2. Procedures

The demographic characteristics of all participants and the medical history of the RA group were collected by interview. Disease activity using the Disease Activity Score-28 joints (DAS-28) [16] and anthropometric measurements (body mass and height) were assessed. The level of physical activity, balance confidence, and functional capacity were recorded from all participants using the International Physical Activity Questionnaire-short form [17], Activities-specific Balance Confidence (ABC) Scale [18], and the Health Assessment Questionnaire (HAQ) [19], respectively. In addition, all participants answered the question: “How many times did you fall last year?”

The Sensory Organization Test was performed on a Smart Balance Master^®^ (NeuroCom International, Inc., Clackamas, OR, USA) force plates system. Vertical forces applied to the support surface were sampled at 100 Hz and used to calculate the center of pressure (COP). The Smart Balance Master system is assembled with a visual surround and a moving support surface that rotates in the sagittal plane in response to participants’ sway (i.e., sway-referenced). For the surface sway-referenced condition, the surface rotates either in a toes-down or toes-up orientation as the participant sways forward and backward, respectively, maintaining a relatively constant ankle joint angle with relation to the surface. For the visual surround sway-referenced condition, as participants swayed forward or backward, the surround rotated in the respective sway direction so that the participant experienced minimal optic flow. Participants experienced three conditions: SOT-1 (open eyes, fixed support surface and surround (visual, vestibular and somatosensory information available); SOT-2 (closed eyes, fixed support surface and surround (no visual information input); SOT-5 (closed eyes, sway-referenced support surface, and fixed surround (no visual information input and somatosensory input inaccurate) (Table 1). For each trial, participants stood barefoot on the force platform facing the visual surround, which enclosed participants on three sides and extended beyond the range of peripheral vision. Participants’ feet were placed in a standardized position (based on height) recommended by the equipment manufacturer. Participants were instructed to keep their arms by their sides, to look straight ahead, and to stand as still as possible during testing. Participants performed three 20 s long trials in each condition. Condition order was randomized.

### 2.3. Data Processing

Raw COP signal was low-pass filtered (fourth-order Butterworth filter, 10 Hz cutoff frequency), demeaned, and rectified. From this processed COP, the average amplitude (COPAP and COPML), mean velocity (COPvelAP and COPvelML), and maximum (PeakCOPAP and PeakCOPML) value were calculated. Customized MatLab scripts (Matlab 2015, Mathworks, Inc., Natick, MA, USA) were utilized to process the COP signals.

### 2.4. Statistical Analysis

The demographic characteristics, clinical aspects, and COP features were described in terms of mean ± standard deviation values or frequency distributions. To compare the demographic and clinical aspects between groups, independent *t*-test or Mann–Whitney’s U-test were applied for parametric and non-parametric data, respectively. Shapiro–Wilk’s test was used to check data distribution, and Levene’s test was used to verify homoscedasticity. To compare the COP variables between groups, and between postural conditions, we have applied Mann-Whitney’s U test, and Wilcoxon’s paired test, respectively. All statistical analyses were performed using the Statistical Package for Social Sciences (SPSS v. 22.0), considering *p* < 0.05.

## 3. Results

Demographic data, as well as clinical and disease characteristics of RA and Control groups, are summarized in Table 2. Both groups comprised Caucasian women with similar mean age and ~19 years mean working time. The RA group presented a lower educational level and had lower-level occupations than CG. In RA, 60.7% had stopped working or were retired due to the disease (3.5 years retirement median time), while 87.6% of CG were either working or only temporarily unemployed.

Most of the RA and Control group members did not consume alcohol or tobacco. They were overweight, showing low physical activity level, and in menopause, not under any hormonal treatment. Menopause onset was ~4 years earlier in the RA group.

Regarding RA’s clinical features, the symptoms’ onset was 17.5 ± 9.6 years, and the diagnosis and treatment were five years later. The RA’s complains were general pain (53.6%), pain with swollen joints (17.9%), difficulty doing daily activities (14.3%), and joint stiffness and less range of motion (10.7%). The C-reactive protein median was 5.9 mgL^−1^. Three quarters of the RA group showed moderate or severe disease activity, 71.3% presented up to seven swollen joints, and 78.5% up to 14 tender joints. Most RA participants (85.7%) were treated with a combination of csDMARDs and anti-inflammatory or analgesic drugs. Most of the RA group had musculoskeletal (92.9%), rheumatological (71.4%), psychiatric (57.1%), ophthalmological (28.6%), and pulmonary (25.0%) conditions. Most of the RA group (85.7%) showed comorbidities (arterial hypertension, 50.0%; psychiatric disorders, 47.1%; metabolic disorders, 35.7%; other rheumatological diseases, 21.4%; and musculoskeletal diseases, 10.7%). The coadjutant treatments were hydrotherapy (42.9%) or more than one type of treatment (14.3%).

Table 3 shows the functional capacity, self-reported falls, and balance confidence for RA and Controls. The RA and Control groups presented different functional capacity, balance confidence, and self-reported number of falls (in the last year). For functional capacity, most Controls (62.5%) did not present disabilities, while most of the RA group (96.5%) had mild to severe disabilities in reaching (42.8%), daily activities (28.6%), gripping (25.0%), rising from sitting (24.3%), and walking (14.3%). Most of the RA group (79.0%) showed moderate to low balance confidence (60.7% had balance confidence score <65%). All Controls showed high balance confidence. While some RA participants (34.7%) have fallen one to 12 times, no Controls have fallen.

Considering the COP variables, for all SOT conditions, COPvelAP were faster for the Control group, while only for SOT-2, COPvelML was faster for the Control; for the other two SOT conditions (1 and 5), COPvelML were similar. Between SOT conditions, for CG and RA, COPvelAP and COPvelML were faster for SOT-5 than SOT-1, while SOT-1 and SOT-2 presented similar COPvelAP and COPvelML.

For SOT-2 and SOT-5, PeakCOPAP and PeakCOPML were larger for the RA group. The RA group showed the largest PeakCOPAP. For both groups, SOT-1 presented the smallest PeakCOPAP and PeakCOPML, and SOT-5 showed the largest PeakCOPAP and PeakCOPML (Table 4).

## 4. Discussion

This study evaluated the postural sway of people with RA. Postural control is impaired in people with RA [5,6]. Under RA conditions, our results indicate that postural sway increases and is faster when sensory information relies on the somatosensory system. Since the burden of RA is not only physiological, but also impacts postural control, our results improve the understanding of why RA impairs daily and social activities.

People with RA showed greater postural sway in different standing posture conditions. Postural sway increased, compared to the control condition, in all SOT evaluations. To eliminate the influence of vision, we tested the balance of people with RA with eyes closed in various somatosensory information situations. Thus, RA presented the largest postural sway for both AP and ML directions. The maximum value of postural sway may indicate the risk of losing balance control [20], because COP is closer to the edge of the stability limit. Greater postural sway was observed in people with RA [21,22] in the AP and ML directions.

Support instability increases postural sway in people with RA. When the basis of support was unstable, people with RA had greater postural sway compared to the control condition. The inflammation caused by RA [23] can be associated with pain and impair the ability to control balance, increasing the risk of falls. King et al. (2012) [23] did not confirm the hypothesis that the vestibular system is altered in RA, despite showing that people with RA have vestibular alterations and that balance alterations are associated. Therefore, as postural control performs better when the availability and quality of sensory information are greater, the experimental restriction of sensory information exposes the difficulty that people with RA must integrate what is seasonally available to maintain balance. Our results suggest how vision is important for people with RA to control postural sway and maintain stability [24,25]. Balance control depends on joint stability, and this is important for the performance of daily tasks. Our findings showed worse conditions in the AR group in activities of daily living according to the HAQ tests, as well as worse pain conditions and joint limitation in the DAS-28 tests (Table 2 and Table 3). In RA, movement difficulties, joint changes, and neuromuscular dysfunction [6,26] compromise the maintenance of balance, increasing the risk of falls. When the base of support is unstable, coordination between joints must be adequate to compensate for unexpected movements of the base of support, which is the condition of SOT-5.

People with RA presented slower postural sway. RA is associated with longer reaction time and movement time [14], which may induce delays in reactive responses in dynamic balance tests. To adapt this condition, COP migration needs to slow down. In addition, the systemic inflammation of RA can deform joint surfaces and reduce joint mobility. These neuromotor and anatomical functional changes may increase the risk of falls in people with RA. In addition, pain associated with systemic inflammation can reduce the practice of physical activity and the development of physical fitness, reducing functional capacity, leading to joint problems linked to muscle atrophy and disuse, and impairing the proper use of postural strategies [22,27]. Thus, slowly moving the COP seems to be an adaption to such impairments.

Sensory restriction increases postural sway velocity for both groups. Standing with eyes closed on a stable or unstable base increases the speed of postural sway in the frontal and sagittal planes. Our participants had a mild to moderate disease index. Therefore, a higher RA level is associated with more falls, less joint mobility, more fear of falling, and greater COP displacement [1,6,28]. In our study, people with RA showed a loss of functionality and low confidence in balance.

Dynamic balance assessments, such as the SOT test, are important tools to understand postural control in RA. Aydoğ et al. (2006) [5] showed that performance in dynamic tests of balance and functional capacity worsened with disease activity. Maintaining an upright posture in static and dynamic conditions, with different availability of sensory information, helps to understand the greater postural fragility in this population, and the eventual increase in the risk of falls. A limitation of this study is to evaluate only reactive responses of postural control. Furthermore, we did not assess how it is possible to improve postural control in people with RA, as customizing physical activity programs for people with RA can improve physical conditions and motor perception [11,26].

## 5. Conclusions

This study showed that people with RA presented larger and slower postural sway during SOT tasks. This was evidenced when visual information was absent. Changes in the postural dynamic responses can be associated with impairments in motor control and cause falls in people with RA.

## Figures and Tables

**Table 1 ijerph-20-04702-t001:** Visual surround and support surface conditions for the Sensory Organization Test (SOT). Sway-referencing involves an anterior/posterior rotation of the platform and/or visual surround that occurs as a response to the person’s shifts in center of pressure.

Condition	Visual Surround	Support Surface	Eyes
SOT-1	Fixed	Fixed	Open
SOT-2	Fixed	Fixed	Closed
SOT-5	Fixed	Sway-referenced	Closed

**Table 2 ijerph-20-04702-t002:** Summary of demographic, clinical, and disease features.

Variables		RA (*n* = 28)	CG (*n* = 16)	Total (*n* = 44)	*p*-Value
		** *Mean ± SD* **			
Age (year)		57.4 ± 9.0	58.5 ± 6.6	57.8 ± 8.1	0.32 ^a^
Working time (year)		21.8 ± 13.0	15.8 ± 13.07	19.6 ± 13.2	0.07 ^a^
		*Median (95%CI)*			
Retirement time/period out of work due to health problems (year)		3.50 (3.29–8.35)	0.00 (−0.22–1.47)	0.00 (2.15–5.71)	**0.001 ^b^**
Social status (minimum wages/month)		3.00 (2.55–5.13)	3.00 (2.37–6.69)	3.00 (3.00–5.18)	0.28 ^b^
		*Absolute and relative frequencies*			
Ethnicity *	White	21 (77.8)	15 (93.8)	36 (83.7)	0.08 ^b^
	Black	4 (14.8)	1 (6.3)	5 (11.6)	
	Mulatto	2 (7.4)	-	2 (4.7)	
Professional status	Employed	11 (39.3)	13 (81.3)	24 (53.5)	**0.01 ^b^**
	Stopped working for health reasons	2 (7.1)	0 (0.0)	2 (4.5)	
	Retired for health reasons	15 (53.6)	2 (12.5)	17 (38.6)	
	Unemployed	-	1 (6.3)	1 (2.3)	
Marital Status	Single	4 (14.3)	1 (6.3)	5 (11.4)	0.14 ^b^
	Married	18 (64.3)	10 (62.5)	28 (63.6)	
	Divorced	5 (17.9)	3 (18.8)	8 (18.2)	
	Widowed	1 (3.6)	1 (6.3)	2 (4.5)	
	Living with a partner	-	1 (6.3)	1 (2.3)	
Educational level	Primary (incomplete)	9 (32.1)	1 (6.3)	10 (22.7)	**0.01 ^b^**
	Primary (complete)	4 (14.3)	-	4 (9.1)	
	Secondary (incomplete)	2 (7.1)	1 (6.3)	3 (6.8)	
	Secondary (complete)	3 (10.7)	6 (37.5)	9 (20.5)	
	University degree (incomplete)	1 (3.6)	-	1 (2.3)	
	University degree (complete)	9 (32.1)	8 (50.0)	17 (38.6)	
Profession	Steward/Receptionist/Businessman	4 (14.3)	4 (25.0)	8 (18.2)	0.25 ^b^
	Registrar/Legal advisor/Journalist/	1 (3.6)	-	1(2.3)	
	Student	-	1 (6.3)	1 (2.3)	
	Housekeeper/Governess/Washerwoman/Ironing clothes work/Kitchen assistant/Housewives	12 (42.9)	1 (6.3)	13 (29.5)	
	Seamstress/Fashion designer/Costume designer	2 (7.1)	1 (6.3)	3 (6.8)	
	Receiver/Handyman/Caretaker/Watchman	1 (3.6)	3 (18.8)	4 (9.1)	
	Teacher/Occupational counselor	4 (14.3)	2 (12.5)	6 (13.6)	
	Maintenance technician (mechanic)	2 (7.1)	-	2 (4.5)	
	Nursing or Dentistry assistant/Physiotherapist/Physical educator	-	3 (18.8)	3 (6.8)	
	Manicure/Hairdresser	1 (3.6)	-	1 (2.3)	
	Geographer	1 (3.6)	-	1 (2.3)	
	Engineer	-	1 (6.3)	1 (2.3)	
Professional status	Employed	11 (39.3)	13 (81.3)	24 (54.5)	**0.01 ^b^**
	Leaved from work for health reasons	2 (7.1)	-	2 (4.5)	
	Retired for health reasons	15 (53.6)	2 (12.5)	17 (38.6)	
	Unemployed	-	1 (6.3)	1 (2.3)	
Physical activity level	Low	13 (46.4)	7 (43.8)	20 (45.5)	0.48 ^b^
	Moderate	9 (32.1)	6 (37.5)	15 (34.1)	
	High	6 (21.4)	3 (18.8)	9 (20.5)	
		*Median (95%CI)*			
	METS (minute/week score)	1350.0 (1229.4–3179.7)	1387.5 (905.5–2752.9)	1383.0 (1382.0–2776.8)	0.50 ^b^
		*Mean ± SD*			
Body mass (kg)		75.88 ± 16.31	70.86 ± 11.17	74.06 ± 14.71	0.14 ^a^
Height (m)		1.61 ± 0.07	1.60 ± 0.07	1.61 ± 0.07	0.34 ^a^
Body mass index (kg/m^2^)		29.17 ± 5.29	27.52 ± 2.87	28.57 ± 4.59	0.09 ^a^
Menopausal age (year)		44.04 ± 5.14	48.38 ± 5.72	45.61 ± 5.68	**0.01 ^a^**
Onset of the symptoms (year)		17.50 ± 9.66	-	-	-
Time to diagnosis (year)		12.87 ± 8.84	-	-	-
Treatment time (year)		12.69 ± 8.91	-	-	-
		*Absolute and relative frequencies*			
Menopause/hormone treatment	No/No	5 (17.9)	3 (18.8)	8 (18.2)	0.24 ^b^
	No/Yes	-	-	-	
	Yes/No	18 (64.3)	12 (75.0)	30 (68.2)	
	Yes/Yes	5 (17.9)	1 (6.3)	6 (13.6)	
Tobacco use	Yes	2 (7.1)	1 (6.3)	3 (6.8)	0.22 ^b^
	No	19 (67.9)	13 (81.3)	32 (72.7)	
	Progress	7 (25.0)	2 (12.5)	9 (20.5)	
Alcohol use	Yes	2 (7.1)	5 (31.3)	7 (15.9)	0.10 ^b^
	No	24 (85.7)	9 (56.3)	33 (75.0)	
	Progress	2 (7.1)	2 (12.5)	4 (9.1)	
Disease activity (DAS-28)	Low (DAS-28 score < 3.2)	6 (21.4)	-	6 (21.4)	-
	Moderate (3.2 ≤ DAS-28 ≤ 5.1)	15 (53.6)	-	15 (53.6)	
	High (DAS-28 > 5.1)	6 (21.4)	-	6 (21.4)	
		*Median (95%CI)*			
C-reactive Protein (mg/L)		5.90 (3.63–7.31)	-	5.90 (3.63–7.31)	-
Swollen joints	0–7	20 (71.3)	-	20 (71.3)	-
	8–14	4 (14.4)	-	4 (14.4)	
	15–22	2 (7.2)	-	2 (7.2)	
	23–28	2 (7.2)	-	2 (7.2)	
Tender joints	0–7	17 (60.7)	-	17 (60.7)	-
	8–14	5 (17.8)	-	5 (17.8)	
	15–22	4 (14.4)	-	4 (14.4)	
	23–28	2 (7.2)	-	2 (7.2)	
Pharmacological treatments	Only inflammatory or analgesic drugs	1 (3.6)	-	-	-
	csDMARDS	1 (3.6)	-	-	
	bDEMARDS	3 (10.7)	-	-	
	Mixed	23 (85.7)	-	-	
	No treatment	-	-	-	
Coadjuvant treatment	Physiotherapy	-	1 (6.3)	1 (2.3)	0.40 ^b^
	Massage therapy	1 (3.6)	-	1(2.3)	
	Hydrotherapy	12 (42.9)	-	12 (27.3)	
	Acupuncture	1 (3.6)	-	1 (2.3)	
	Mixed	4 (14.3)	2 (12.5)	6 (13.6)	
	No treatment	10 (35.7)	13 (81.3)	23 (52.3)	
Family history of rheumatic diseases	Yes	19 (67.9)	6 (37.5)	25 (56.8)	**0.04 ^b^**
	No	7 (25.0)	9 (56.3)	16 (36.4)	
	Unknown	2 (7.1)	1 (6.3)	3 (6.8)	
Type of rheumatic diseases in family	RA	10 (50.0)	1 (14.3)	11 (40.7)	0.15 ^b^
	Other	3 (15.0)	3 (42.9)	6 (22.2)	
	Unknown	7 (35.0)	3 (42.9)	10 (37.0)	
Conditions associated with RA	Present	26 (92.9)	11 (68.8)	37 (84.1)	**0.01 ^b^**
	Absent	2 (7.1)	5 (31.3)	7 (15.9)	
Manifestations associated with RA	Pulmonary	7 (25.0)	-	7 (15.9)	**0.01 ^b^**
	Ocular	8 (28.6)	-	8 (18.2)	**0.009 ^b^**
	Cardiac	4 (14.3)	-	4 (9.1)	0.05 ^b^
	Neurological	4 (14.3)	1 (6.3)	5 (11.4)	0.21 ^b^
	Rheumatological	20 (71.4)	-	20 (45.5)	**<0.001 ^b^**
	Psychiatric	16 (57.1)	-	16 (36.4)	**<0.001 ^b^**
	Haematological	5 (17.9)	2 (12.5)	7 (15.9)	0.32 ^b^
	Musculoskeletal	26 (92.9)	3 (18.8)	29 (65.9)	**<0.001 ^b^**
	No manifestations	-	11 (68.8)	11 (25.0)	**<0.001 ^b^**
Presence of comorbidities	Yes	24 (85.7)	9 (56.3)	33 (75.0)	**0.01 ^b^**
	No	4 (14.3)	7 (43.8)	11 (25.0)	
Type of comorbidities	Rheumatological				
	Sjögren-Larsson syndrome	1 (3.6)	-	1 (2.3)	0.22 ^b^
	Lupus erythematosus systemic	2 (7.1)	-	2 (4.5)	0.13 ^b^
	Fibromyalgia	3 (10.7)	-	3 (6.8)	0.09 ^b^
	Tuberculosis	1 (3.6)	-	1 (2.3)	0.22 ^b^
	Metabolic				
	Dyslipidemia	4 (14.3)	3 (18.8)	7 (15.9)	0.35 ^b^
	Diabetes mellitus	6 (21.4)	-	6 (13.6)	**0.02 ^b^**
	Thyroid dysfunctions	5 (17.9)	-	5(11.4)	**0.03 ^b^**
	Cardiac				
	Mitral valve disease/Atrial fibrillation	1 (3.6)	-	1 (2.3)	0.22 ^b^
	Vascular diseases	2 (7.1)	1 (6.3)	3 (6.8)	0.455 ^b^
	Hypertension	14 (50.0)	4 (25.0)	18 (40.9)	0.05 ^b^
	Chronic obstructive pulmonary disease	1 (3.6)	-	1 (2.3)	0.22 ^b^
	Psychiatric		-	-	
	Depression	8 (28.6)	-	8 (18.2)	**0.009 ^b^**
	Other psychiatric disorders	5 (18.5)	-	5 (11.6)	**0.03 ^b^**
	Musculoskeletal		-	-	
	Osteoporosis/Osteopenia	1 (3.6)	1 (6.3)	2 (4.5)	0.34 ^b^
	Osteoarthrosis	2 (7.1)	1 (6.3)	3 (6.8)	0.45 ^b^
Main complaints	Pain	15 (53.6)	-	-	-
	Pain, joint stiffness/ROM diminished	3 (10.7)	-	-	
	Pain and swollen joints	5 (17.9)	-	-	
	Pain and difficulty doing daily activities	4 (14.3)	-	-	
	No complaints	1 (3.6)	-	-	

RA: rheumatoid arthritis group; CG: Control group. SD: Standard deviation. 95%CI: 95% Confidence interval. * Brazil’s ethnic classification according to the Brazilian Institute of Geography and Statistics. csDEMARDS conventional synthetic disease modifying antirheumatic drugs (Methotrexate, Sulfasalazine, Leflunomide, Hydroxy-chloroquine or MMF or azathioprine); bDEMARD biologic disease modifying antirheumatic drugs (TNF inhibitor, Abatacept, Rituximab, Tocilizumab or Anakinra). ROM range of motion. Frequencies (%) were calculated in relation to the group (RA, GC or total). ^a^
*p*-value (one sided) calculated for the independent *t*-test. ^b^
*p*-value (one sided) calculated for Mann–Whitney U test. Bold: *p* < 0.05.

**Table 3 ijerph-20-04702-t003:** Functional capacity, self-reported number of falls, and balance confidence results.

Variables		RA (*n* = 28)	CG (*n* = 16)	Total (*n* = 44)	*p*-Value
		*Absolute and relative frequencies*			
Functional capacity (HAQ)	0 (no disability)	1 (3.6)	10 (62.5)	11 (25.0)	**<0.001 ^b^**
	0.1 to 1 (mild to moderate disability)	15 (53.6)	5 (31.3)	20 (45.5)	
	>1 to 2 (moderate to severe disability)	12 (42.9)	-	12 (27.3)	
	>2 to 3 (severe to very severe disability)	-	-	-	
HAQ sections					
Dressing	0 (without any difficulty)	13 (46.4)	15 (93.8)	28 (63.6)	**0.001 ^b^**
	1 (with some difficulty)	11 (39.3)	1 (6.3)	12 (27.3)	
	2 (with much difficulty)	4 (14.3)	-	4 (9.1)	
	3 (unable to do)	-	-	-	
Arising	0 (without any difficulty)	10 (35.7)	14 (87.5)	24 (54.5)	**<0.001 ^b^**
	1 (with some difficulty)	12 (42.9)	2 (12.5)	14 (31.8)	
	2 (with much difficulty)	4 (17.2)	-	4 (9.1)	
	3 (unable to do)	2 (7.1)	0	2 (4.5)	
Eating	0 (without any difficulty)	9 (32.1)	15 (93.8)	24 (54.5)	**<0.001 ^b^**
	1 (with some difficulty)	16 (57.1)	1 (6.3)	17 (38.6)	
	2 (with much difficulty)	3 (10.7)	-	3 (6.8)	
	3 (unable to do)	-	-	-	
Walking	0 (without any difficulty)	9 (32.1)	15 (93.8)	24 (54.5)	**<0.001 ^b^**
	1 (with some difficulty)	15 (53.6)	1 (6.3)	16 (36.4)	
	2 (with much difficulty)	3 (10.7)	-	3 (6.8)	
	3 (unable to do)	1 (3.6)	-	1 (2.3)	
Hygiene	0 (without any difficulty)	14 (50.0)	16 (100.0)	30 (68.2)	**<0.001 ^b^**
	1 (with some difficulty)	12 (42.9)	-	12 (27.3)	
	2 (with much difficulty)	2 (7.1)	-	2 (4.5)	
	3 (unable to do)	-	-	-	
Reach	0 (without any difficulty)	6 (21.4)	14 (87.5)	20 (45.5)	**<0.001 ^b^**
	1 (with some difficulty)	10 (35.7)	2 (12.5)	12 (27.3)	
	2 (with much difficulty)	9 (32.1)	-	9 (20.5)	
	3 (unable to do)	3 (10.7)	-	3 (6.8)	
Grip	0 (without any difficulty)	12 (42.9)	16 (100.0)	28 (63.6)	**<0.001 ^b^**
	1 (with some difficulty)	9 (32.1)	-	9 (20.5)	
	2 (with much difficulty)	5 (17.9)	-	5 (11.4)	
	3 (unable to do)	1 (7.1)	-	1 (4.5)	
Common daily activities	0 (without any difficulty)	8 (28.6)	14 (87.5)	22 (50.0)	**<0.001 ^b^**
	1 (with some difficulty)	12 (42.9)	2 (12.5)	14 (31.8)	
	2 (with much difficulty)	7 (25.0)	-	7 (15.9)	
	3 (unable to do)	1 (3.6)	-	1 (2.3)	
Self-reported number of falls in the last year	0	18 (64.3)	16 (100.0)	34 (77.3)	**0.004 ^b^**
	1 to 6	9 (32.1)	-	9 (20.4)	
	12	1 (3.6)	-	1 (2.3)	
Balance confidence (ABC)	ABC score < 50 (low)	4 (14.4)	-	4 (9.2)	**<0.001 ^b^**
	50 ABC score < 80 (%) (moderate)	18 (64.6)	-	18 (41.22)	
	ABC score 80 (high)	6 (21.6)	16 (100.0)	22 (50.0)	

RA rheumatoid arthritis group; CG Control group. SD Standard deviation. HAQ Health Assessment Questionnaire. ABC Activities Specific Balance Confidence Scale. ^b^ *p*-value (one sided) calculated for the Mann–Whitney U test. Bold: significant for *p* < 0.05.

**Table 4 ijerph-20-04702-t004:** Effect of the groups (RA or CG) and of the SOT conditions on the biomechanical parameters.

Group	Variable	Condition (Median (95%CI))		
		SOT1	SOT2	SOT5
RA (*n* = 28)	COPvelml (cm/s)	0.49 (0.49–0.64) ^y^	0.82 (0.66–0.84) ^x,y^	3.88 (3.90–5.80) ^y^
	PeakCOPml (cm)	0.27 (0.28–0.40) ^y^	0.39 (0.41–0.55) ^x,y^	0.89 (0.87–1.08) ^x,y^
	COPvelap (cm/s)	0.10 (0.09–0.10) ^x,y^	0.10 (0.09–0.11) ^x^	0.20 (0.19–0.25) ^x,y^
	PeakCOPap (cm)	0.82 (0.78–0.93) ^x,y^	1.40 (1.40–1.73) ^x,y^	0.69 (0.66–0.81) ^x,y^
CG (*n* = 16)	COPvelml (cm/s)	0.65 (0.58–0.86) ^y^	1.14 (0.95–1.33) ^x,y^	4.74 (4.16–6.63) ^y^
	PeakCOPml (cm)	0.23 (0.22–0.39) ^y^	0.25 (0.25–0.30) ^x^	0.89 (0.87–1.08) ^x,y^
	COPvelap (cm/s)	0.12 (0.12–0.14) ^x,y^	0.14 (0.13–0.17) ^x^	0.39 (0.33–0.65) ^x,y^
	PeakCOPap (cm)	0.53 (0.55–0.71) ^x,y^	0.84 (0.76–0.96) ^x,y^	0.69 (0.66–0.81) ^x,y^

RA rheumatoid arthritis group; CG Control group. 95%CI: 95% Confidence interval. Superscript ‘x’ means statistically significant difference (*p* < 0.05, one sided) between RA and CG by Mann–Whitney’s U test. Superscript ‘y’ means statistically significant difference (*p* < 0.05, one sided) in comparison with SOT1 condition obtained by Wilcoxon paired samples test.

## Data Availability

Data can be available for further analysis under authors’ approval.

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
