# Peer review of "Sensory Integration for Postural Control in Rheumatoid Arthritis Revealed by Computerized Dynamic Posturography"

_ijerph, 2023, doi:10.3390/ijerph20064702_

Round 1

Reviewer 1 Report

The authors investigated sensorimotor integration for postural control of 28 women with RA compared to 16 women in the group control. They presented an elaborated characterization of the subjects with demographic and clinical variables and measured COP ML and AP velocity during conditions 1, 2, and 5 of the sensory organization test. Postural control in RA is a topic of utmost importance to the field as fear of falling can reach to almost 50% of patients with RA, and ˜20% this population experienced a fall episode in the past year. Results from this study have the potential to inform clinical interventions to increase postural stability during functional activities and to prevent falls.  However, I noticed an important typo on Table 3 that contains the main results of this article.  Results for RA and CG groups are exactly the same for SOT5, and results for COP ML and AP velocity are the same within groups for the other two conditions. This seems very unlikely. I am happy to revisit this article with the correct results and sections rewritten.

Author Response

Dear Reviewer,

Thank you for advising us about table 3. We reviewed the data analysis to add the correct data to this table. Thus, we have also changed the results and discussion sections. Corrections are painted in yellow.

Reviewer 2 Report

This reviewer appreciates the time and effort invested by the authors in conducting the study and writing the manuscript. This reviewer has following questions/opinions regarding the study, as is presented in the manuscript:

1. With regard to the following sentence on line 71-72

"In addition, all participants an-71 swered the question: "How many times did you fall last year?" "

Did the authors collect any information on what led to the fall? did the participants fell unexpectedly or in a genuine circumstance where anyone could fall?

2. Were there any patients with flat feet? Was flat feet a parameter used for excluding participants from the two patient groups? If so, that must be mentioned in the methods section.

3. This reviewer feels that the participant size is too small to arrive at any definitive conclusions. So the authors must either justify how the number of participants are sufficient for them to arrive at their conclusions or they must include the limited number of patients among the limitation of their study. 

Author Response

Dear Reviewer,

Thank you for your comments. Each comment is replied to below.

1. With regard to the following sentence on lines 71-72 "In addition, all participants an-71 answered the question: "How many times did you fall last year?" "Did the authors collect any information on what led to the fall? did the participants fall unexpectedly or in a genuine circumstance where anyone could fall?

Answer: No, we did not. We have just asked how many times they fell, without asking for detailed information about how they fell.

2. Were there any patients with flat feet? Was flat feet a parameter used for excluding participants from the two patient groups? If so, that must be mentioned in the methods section.

Answer: No, we did not evaluate the participants’ feet. Feet conditions can influence balance control, but we did not perform any evaluation of foot sensibility, muscle strength, or mobility.

3. This reviewer feels that the participant size is too small to arrive at any definitive conclusions. So, the authors must either justify how the number of participants is sufficient for them to arrive at their conclusions or they must include the limited number of patients among the limitation of their study.

Answer: Unfortunately, we are not able to add more participants to this study. Therefore, we have added this sentence in the last paragraph of the discussion section: An important limitation in our study is the sample size, thus our results should be carefully applied and a similar study should be replicated with a larger sample size.

Round 2

Reviewer 2 Report

All of this reviewer's comments have been addressed satisfactorily in the revised version.

The manuscript can be considered for publication.